

# Review article: Wave analysis methods for space plasma experiment

Yasuhito Narita[1]

[1]Space Research Institute, Austrian Academy of Sciences, Schmiedlstr. 6, A-8042 Graz, Austria

*Correspondence to:* Y. Narita
(yasuhito.narita@oeaw.ac.at)

**Abstract.** A review of analysis methods is given on quasi-monochromatic waves, turbulent fluctuations, wave-wave and wave-particle interactions for single-spacecraft data in situ in near-Earth space and interplanetary space, in particular using magnetic field and electric field data. Energy spectra for different components of the fluctuating fields, minimum variance analysis, propagation and polarization properties of electromagnetic waves, wave distribution function, helicity quantities, higher-order statistics, and detection methods for wave-particle interactions are explained.

## 1 Introduction

### 1.1 Turbulent space plasma

Waves and turbulence phenomena are observed in various regions of near-Earth space and interplanetary space such as in the solar wind, the foreshock, and the magnetosheath. Turbulence plays an important role in solar plasma (coronal heating and dynamo mechanism), collisionless shocks (particle acceleration), and interstellar space (diffusion of galactic cosmic ray). Due to its electrically conducting nature and collisionless nature, the picture of energy cascade of plasma turbulence is more diverse than that of fluid turbulence. Plasma physical processes such as wave-wave and wave-particle interactions serve as a channel of the energy cascade in addition to eddy splitting intrinsic to the fluid-like behavior of plasma. On kinetic scales of the order of the ion gyro-radius (about 400 km in the solar wind) or the electron gyro-radius (about 10 km), waves become dispersive while interacting with individual particles (particle acceleration and scattering).

This paper is a review of analysis methods for waves, turbulence, wave-wave interactions, and wave-particle interactions using in situ measurement data of magnetic and electric fields. A summary of the analysis methods is displayed in Tab. 1. The emphasis of the review is on the single-spacecraft measurements of the magnetic field and the electric field, primarily using the second-order quantities such as energy and helicity. While many of the current spacecraft missions perform multi-point measurements: four-point tetrahedral formation flight on scales of 10,000 km down to 100 km by the Cluster mission (Escoubet et al., 2001) and on 10 km scale by the MMS mission (Burch et al., 2016), one-dimensional five-point measurements by the THEMIS mission (Angelopoulos, 2008), three-point measurements by the Swarm mission (Chulliat et al., 2013; Olsen et al., 2013; Thebault et al., 2013), and two-point measurements by Van Allen Probes (Mauk et al., 2013; Stratton et al., 2013), the upcoming spacecraft missions are more specialized to unique observational approaches toward the understanding of turbulence processes (particularly in interplanetary space) at the cost of returning back to single spacecraft measurements. Examples are



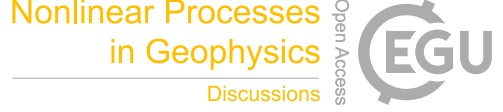

**Table 1.** Wave and turbulence analysis methods.

| target | symbol | input data |
|---|---|---|
| energy spectra | $|B_\perp|^2, |B_\parallel|^2$ | $\boldsymbol{B}$ |
| | $|B_{\mathrm{R}}|^2, |B_{\mathrm{L}}|^2$ | $\boldsymbol{B}$ |
| | $|E_\perp|^2, |E_\parallel|^2$ | $\boldsymbol{E}$ |
| | $|z^+|^2, |z^-|^2$ | $\boldsymbol{B}$ and $\boldsymbol{U}$ |
| compressibility | $|B_\parallel|^2/|\boldsymbol{B}|^2$ | $\boldsymbol{B}$ |
| ellipticity | $\epsilon$ | $\boldsymbol{B}$ or $\boldsymbol{E}$ |
| wavevector angle | $\theta_{\mathrm{kB}}$ | $\boldsymbol{B}$ |
| phase speed | $v_{\mathrm{ph}}$ | $\boldsymbol{B}$ and $\boldsymbol{E}$ |
| Poynting vector | $\boldsymbol{S}$ | $\boldsymbol{B}$ and $\boldsymbol{E}$ |
| helicity quantities | $h_{\mathrm{m}}, h_{\mathrm{c}}$ | $\boldsymbol{B}$ or $\boldsymbol{U}$ |
| wave distribution function | $F(\boldsymbol{k})$ | $\boldsymbol{B}$ or $\boldsymbol{E}$ |
| higher order statistics | $C^{(n)}$ | $\boldsymbol{B}$, $\boldsymbol{E}$, or $n$ |
| resonance parameter | $\zeta_{\mathrm{L}}, \zeta_{\mathrm{C}}$ | $\omega$, $\boldsymbol{k}$, and $v_{\mathrm{th}}$ |
| pitch angle scattering | - | $f(\boldsymbol{v})$ |

simultaneous remote sensing and in situ measurements by Solar Orbiter (Müller et al., 2013), the closest observations to the Sun by Solar Probe Plus (Fox et al., 2015), and high-precision sampling of particle velocity distribution functions, electric field, and magnetic field by THOR (Vaivads et al., 2016). Analysis of spatial structure or intermittency is not covered here, and will be presented in separate papers.

## 1.2 Fluctuation types

Turbulent fields may be composed of various fluctuation types such as linear mode waves, nonlinear wave component, and coherent structures. An overview of these fluctuation types is given here.

**Linear mode waves**

While magnetohydrodynamics (MHD) hosts three distinct linear wave modes (fast, Alfvén, and slow modes), the kinetic treatment of plasma exhibits a larger number of linear mode waves. Some are natural extensions of the MHD modes, and the others are of purely kinetic origin, resulting from the wave resonance with individual electrons or ions. Kinetic wave modes from ion to electron scales relevant to plasma turbulence (for oblique propagations to the mean magnetic field) include the whistler mode, the ion Bernstein mode, the kinetic Alfvén mode, the kinetic slow mode, the lower hybrid mode, the electron





cyclotron mode, the electron Bernstein mode, and the upper hybrid mode (for a Maxwellian plasma). The dispersion relations

40   are schematically shown in Fig. 1.

–  Whistler mode is an extension of the MHD fast mode to the kinetic scales (from the ion gyro-scale down to the electron gyro-scale) (Gary, 1986). Whistler mode is a right-hand circularly polarized mode (with the rotation sense of electron gyration), and can exist even in the limit to perpendicular propagation. The whistler mode extends to the electron cyclotron mode at higher frequencies.

45   –  Ion Bernstein mode is of strongly electrostatic nature, and appears as a series of resonance break-up of the whistler mode at the harmonics of the ion cyclotron frequency in the limit to perpendicular propagation. Since the ion Bernstein mode has higher frequencies than the ion cyclotron frequency, the ion Bernstein mode can serve as "stations" of wave-wave couplings and can sustain the daughter waves for a longer time, enabling a cascade of the fluctuation energy to higher frequencies (Jenkins et al., 2013).

–  Kinetic Alfvén mode is a small-scale extension of the MHD Alfvén mode to the ion kinetic domain, and is obtained as a limit to perpendicular propagation of the ion cyclotron mode. The sense of dispersion relation shows a transition at a

propagation angle of $70°$ to $75°$ that the frequencies increase more slowly at higher wavenumbers; The transition is from a convex curvature sense of the dispersion relation for the ion cyclotron mode into a concave sense for the kinetic Alfvén mode.

–  Kinetic slow mode is a counterpart to the kinetic Alfvén mode, and is the ion-kinetic extension of the MHD slow mode. The kinetic slow mode is of highly compressible nature, and is obtained as a quasi-perpendicular limit of the low-

frequency ion acoustic waves. The kinetic slow mode is only moderately damped in the quasi-perpendicular directions (at angles around $85°$ and larger).

–  Lower hybrid mode is of strongly electrostatic nature and is a resonance mode of the gyro-motion of the both electrons and ions. The resonance frequency is about 43 times higher than the proton cyclotron frequency (it is at $\sqrt{m_p/m_e}\Omega_p$). This mode is considered as an efficient heating mechanism because the wave can accelerate electrons through the cy-

clotron resonance parallel to the mean magnetic field and simultaneously ions through the Landau resonance perpendicular to the mean field.

–  Electron cyclotron mode propagates in the parallel to oblique directions to the mean magnetic field. The frequency rises up to the electron cyclotron frequency at which the wave electric field is in resonance with the electron gyration. The resonance frequency becomes lower at larger propagation angles. In the limit to perpendicular propagation, the resonance

frequency falls down onto the lower hybrid frequency.

–  Electron Bernstein mode is of strongly electrostatic nature, and appears at frequencies close to the harmonics of the electron cyclotron frequency.





– Upper hybrid mode is a resonance mode as a result of the coupling of electron gyro-motion with the electron plasma oscillation. The frequency is higher than the electron cyclotron frequency.

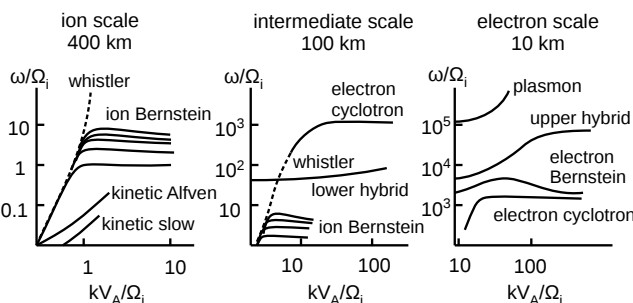

**Figure 1.** Schematic dispersion relations on ion-kinetic to electron-kinetic scales for oblique wave propagation to the mean magnetic field.

### Nonlinear modes

Nonlinear modes can be any propagating wave components other than the linear mode fluctuations. Nonlinear modes may appear as large-amplitude solitary waves or as small-amplitude sideband waves at frequencies around that of the linear mode. The lifetime can be different and presumably depend on the fluctuation amplitude. Solitary waves may be stable if the wave steepening effect is balanced against the dispersion effect. Sideband waves may break into other frequencies and wavevectors through a successive wave-wave interaction.

### Coherent structures

Coherent structures appear in various forms such as eddies, current sheets, flux tubes, density cavities, or shocklets. Flux tubes may be twisted around their axis, which can be deceptive to a circularly rotating wave. Coherent structures are different from waves in that the coherent structures do not propagate intrinsically, and appear as a zero-frequency mode in terms of wave analysis. Formation of thin current sheet leads to a hypothesis of electron gyration-scale magnetic reconnection as an effective diffusion mechanism of turbulent fluctuations (Treumann et al., 2015).

## 2 Quasi-monochromatic waves

### 2.1 Cross spectral density matrix

Quasi-monochromatic waves are identified as local peaks in the energy spectrum. For vectorial quantities such as the magnetic field, the electric field, and the flow velocity, the method of cross spectral density matrix is particularly useful to extract the information on the wave properties.





The measured field data are Fourier transformed from the time domain into the frequency domain:

$$\tilde{\boldsymbol{B}}(\omega) = \int\limits_0^T \mathrm{d}t \, \boldsymbol{B}(t)\mathrm{e}^{\mathrm{i}\omega t}. \tag{1}$$

Here we choose the magnetic field data in the time domain $\boldsymbol{B}(t)$. The Fourier transformed quantity is expressed by tilde, $\tilde{\boldsymbol{B}}(\omega)$.

We use the angular frequencies $\omega$. The dimension of the Fourier transformed quantity is different from the original quantity because of the integration over the time. In the case of Eq. (1), we compute the Fourier coefficients by the integration (i.e., continuous distribution of the spectrum) using the time length $T$. In the case of discrete time series sampling and the discrete Fourier transform, the dimension must be adapted accordingly.

The cross spectral density (CSD) matrix $\mathbf{R}$ is constructed in the frequency domain as:

$$\mathbf{R} = \frac{1}{T}\left\langle \tilde{\boldsymbol{B}}(\omega)\tilde{\boldsymbol{B}}^{\dagger}(\omega) \right\rangle, \tag{2}$$

Here the dagger denotes the operation of Hermitian conjugate, and the angular bracket the operation of statistical averaging. The CSD matrix in the component-wise expression is:

$$\mathbf{R} = \frac{1}{T}\left[ \begin{array}{ccc} \langle\tilde{B}_{\mathrm{x}}\tilde{B}_{\mathrm{x}}^*\rangle & \langle\tilde{B}_{\mathrm{x}}\tilde{B}_{\mathrm{y}}^*\rangle & \langle\tilde{B}_{\mathrm{x}}\tilde{B}_{\mathrm{z}}^*\rangle \\ \langle\tilde{B}_{\mathrm{y}}\tilde{B}_{\mathrm{x}}^*\rangle & \langle\tilde{B}_{\mathrm{y}}\tilde{B}_{\mathrm{y}}^*\rangle & \langle\tilde{B}_{\mathrm{y}}\tilde{B}_{\mathrm{z}}^*\rangle \\ \langle\tilde{B}_{\mathrm{z}}\tilde{B}_{\mathrm{x}}^*\rangle & \langle\tilde{B}_{\mathrm{z}}\tilde{B}_{\mathrm{y}}^*\rangle & \langle\tilde{B}_{\mathrm{z}}\tilde{B}_{\mathrm{z}}^*\rangle \end{array} \right]. \tag{3}$$

The asterisk denotes the complex conjugate. It is worth mentioning that the CSD matrix must be averaged over different re-

alizations, e.g., by chopping the time interval into sub-intervals and averaging the matrix over the sub-intervals. Otherwise the matrix becomes singular in that the determinant is zero. Quasi-stationary fluctuations must be assumed for a proper measurement of the CSD matrix. The dimension of CSD matrix elements are in units of square amplitude per frequency such as $\mathrm{nT}^2/\mathrm{Hz}$ in the case of the magnetic field measurement. The choice of the coordinate system is arbitrary. Convenient choices for the CSD matrix representation are to use a physically relevant direction as a reference, e.g., the mean magnetic field direction,

the wavevector direction, the minimum variance direction, or the flow direction (Tab. 2).

The off-diagonal elements represent covariances of different fluctuation components. In general, one may construct the covariance between one of the field components (e.g., parallel magnetic field fluctuation to the mean field) and the other fluctuation field (e.g., plasma density fluctuation). To simplify the argument, the time factor $1/T$ and the tilde symbol are omitted hereafter.

**Mean-field-aligned system**

The CSD matrix is conveniently analyzed by choosing the mean magnetic field as the primary reference direction (mean-field aligned system, MFA) to determine the wave energy for perpendicular and parallel fluctuations (to the mean magnetic field) and the field rotation sense around the mean field (Fowler et al., 1967; Arthur et al., 1976). The CSD matrix in the MFA system



**Table 2.** Notations for different coordinate systems useful in wave and turbulence analysis: MFA (mean-field aligned), K (wavevector), MVA (minimum variance analysis), and ST (streamwise).

| coordinate system | MFA | K | MVA | ST |
|---|---|---|---|---|
| reference vector | $\boldsymbol{B}_0$ | $\boldsymbol{k}$ | $\boldsymbol{e}_{\lambda 3}$ | $\boldsymbol{U}_0$ |
| x | $\perp 1$ | t1 | $\lambda 1$ | $f1$ |
| y | $\perp 2$ | t2 | $\lambda 2$ | $f2$ |
| z | $\parallel$ | $\ell$ | $\lambda 3$ | $fl$ |

is expressed as

$$\mathbf{R}_{\mathrm{MFA}} = \begin{bmatrix} R_{\perp 1 \perp 1} & R_{\perp 1 \perp 2} & R_{\perp 1 \parallel} \\ R_{\perp 2 \perp 1} & R_{\perp 2 \perp 2} & R_{\perp 2 \parallel} \\ R_{\parallel \perp 1} & R_{\parallel \perp 2} & R_{\parallel \parallel} \end{bmatrix}. \tag{4}$$

The secondary reference direction must be specified by choosing, e.g., the Earth-to-Sun direction, the flow direction, or the maximum variance direction in the perpendicular plane. The diagonal elements represent the energies for the perpendicular fluctuating field ($R_{\perp 1 \perp 1} + R_{\perp 2 \perp 2}$, incompressible sense of fluctuation) and that for the parallel fluctuation ($R_{\parallel \parallel}$, compressible sense of fluctuation). One may then plot the respective energies in the spectral domain as in Fig. 2 (typically in the spacecraft-

frame frequency domain). The trace of the CSD matrix $\mathrm{tr}\mathbf{R}$ represents the total fluctuation energy. One may construct the ratio of the compressible fluctuation energy to the total fluctuation energy, $R_{\parallel \parallel}/\mathrm{tr}\mathbf{R}$, as an index of magnetic compressibility.

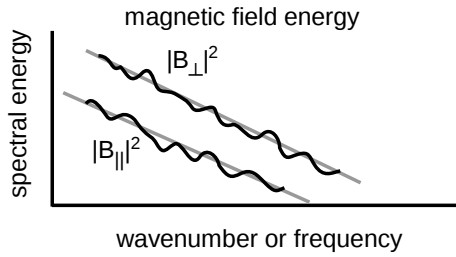

**Figure 2.** Magnetic field energy spectra for parallel and perpendicular fluctuating components to the mean magnetic field.

Rotation sense of the field fluctuation is evaluated from the off-diagonal elements of the CSD matrix. Ellipticity is a useful concept to estimate the polarization, and is defined as the ratio of the minor semi-axis to the major semi-axis of the elliptically rotating field:

$$\epsilon = \frac{B_{\mathrm{minor}}}{B_{\mathrm{major}}} = \tan \psi. \tag{5}$$





The value of ellipticity is bound between $-1$ and $+1$ for left-hand circular polarization and right-hand one, respectively. $\epsilon > 0$ at a positive frequency means the right-hand field rotation sense when viewing into the direction of the mean magnetic field (the same sense as electron gyration), while $\epsilon < 0$ the left-hand field rotation sense. Our definition of the ellipticity follows that in plasma physics (Stix, 1992; Gary, 1993). Another choice for the field rotation sense is to analyze the rotation sense around the wave propagation direction (Born and Wolf, 1980), associated with the notion of the wave helicity. Rotation sense of the wave field can be visualized (and hence double-checked) by plotting the trajectory of the wave field time-evolution in

the perpendicular plane (referred to as the hodogram).

Ellipticity is evaluated through the angle $\psi$ spanned by the minor and the major semi-axes, and the angle $\psi$ is determined by the analysis of the two-by-two CSD sub-matrix of in the $\perp 1$-$\perp 2$ plane, $\mathbf{R}'$,

$$\mathbf{R}' = \begin{bmatrix} R_{\perp 1 \perp 1} & R_{\perp 1 \perp 2} \\ R_{\perp 2 \perp 1} & R_{\perp 2 \perp 2} \end{bmatrix}, \tag{6}$$

through the following relation:

$$\sin(2\psi) = \frac{2\mathrm{Im}\left(R'_{\perp 1 \perp 2}\right)}{\left[\left(\mathrm{tr}\mathbf{R}'\right)^2 - 4\det\left(\mathbf{R}'\right)\right]^{1/2}}. \tag{7}$$

Eq. (7) is derived by modeling the quasi-monochromatic wave as elliptically polarized:

$$\delta\boldsymbol{B} = \begin{bmatrix} B_{\mathrm{major}}\,\mathrm{e}^{i\omega t} \\ B_{\mathrm{minor}}\,\mathrm{e}^{i(\omega t - \pi/2)} \\ 0 \end{bmatrix}, \tag{8}$$

The CSD matrix is then evaluated as

$$\mathbf{R}_{\mathrm{model}} = \begin{bmatrix} B_{\mathrm{major}}^2 & iB_{\mathrm{major}}B_{\mathrm{minor}} & 0 \\ -iB_{\mathrm{major}}B_{\mathrm{minor}} & B_{\mathrm{minor}}^2 & 0 \\ 0 & 0 & 0 \end{bmatrix}. \tag{9}$$

**Minimum variance system**

The CSD matrix can be transformed into a diagonal form by using a unitary matrix, and the wave properties are analyzed in the minimum variance system:

$$\mathbf{R}_{\mathrm{MV}} = \begin{pmatrix} \lambda_1 & 0 & 0 \\ 0 & \lambda_2 & 0 \\ 0 & 0 & \lambda_3 \end{pmatrix}, \tag{10}$$

The diagonal elements contain the largest eigenvalue $\lambda_1$ (maximum variance). the intermediate one $\lambda_2$, and the smallest one

$\lambda_3$ (minimum variance). The associated eigenvectors are orthogonal to one another (because the CSD matrix is hermitian symmetric), pointing the directions of the maximum, intermediate, and minimum variances (Fig. 3).





The essence of the minimum variance analysis lies in the fact that the minimum variance direction $e_{\lambda 3}$ reasonably agrees with the wavevector direction (the longitudinal direction), i.e., $k \cdot B_1 = 0$ (where $B_1$ denotes the fluctuation direction) if the polarization plane spanned by two transverse fluctuating components is well determined, characterized by $\lambda_2 \gg \lambda_3$ (Sonnerup

and Cahill, 1967). The magnetic field is divergence-free ($\nabla \cdot B = 0$), and the fluctuation appears only in the transverse directions (which can be checked by modeling the magnetic field as a composition of the mean field and a monochromatic wave as $B = B_0 + B_1 \mathrm{e}^{\mathrm{i}k \cdot r}$). Note that only the axis of the propagation direction is determined and there still remains a $180°$-ambiguity in the sense of propagation. For a linearly polarized wave, the method of minimum variance analysis does not work because the polarization plane is not uniquely determined. To unambiguously determine the wavevector, different methods need to be combined, e.g., Poynting vector, wave distribution function, or multi-probe methods.

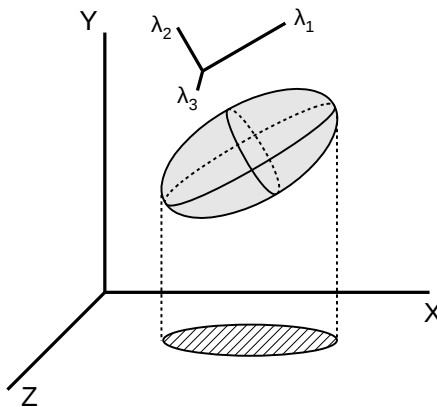

**Figure 3.** Polarization ellipsoid in the minimum variance analysis


## 2.2 Electromagnetic waves

For quasi-monochromatic electromagnetic waves, one may estimate the phase speeds and the wavevectors from the electric and magnetic field data. The phase speed is obtained as a ratio of the electric field amplitude to that of the magnetic field

$$v_{\mathrm{ph}} = \frac{\omega}{k} = \frac{\delta E_{\mathrm{t}1}}{\delta B_{\mathrm{t}2}}. \tag{11}$$

Eq. (11) is obtained from the induction equation $\partial_t B = -\nabla \times E$ for a plane wave. The electric field, the magnetic field, and the propagation direction (wavevector direction) are mutually orthogonal, and one must use the transverse-1 component for the electric field and the transverse-2 component for the magnetic field in Eq. (11). The phase speed is expressed in the observer's frame of reference, and is subject to the Doppler shift in the plasma flow of observer's motion. If multiple waves are present at the same frequency, the phase speed cannot be determined properly.

From the phase speed $v_{\mathrm{ph}}$ as a function of the frequencies, one may obtain the wavenumber as $k = \omega / v_{\mathrm{ph}}$ and therefore the wavenumber-frequency diagram (the dispersion relation diagram, Fig. 4). The wavevector direction (which is accessible in





the minimum variance analysis for elliptically polarized magnetic field fluctuations) can be double-checked with the Poynting vector, $\boldsymbol{S} = \frac{1}{\mu_0}\boldsymbol{E} \times \boldsymbol{B}$.

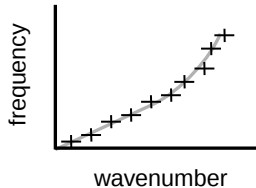

**Figure 4.** Wavenumber-frequency diagram derived from the phase speed estimate.

### 2.3 Wave distribution function

Wave distribution function is the concept of the wave energy distribution in the wavevector domain assuming the existence of dispersion relations (Fig. 5). The analysis for the wave distribution function can be implemented to single spacecraft measurements (Storey and Lefeuvbre, 1979, 1980; Lefeuvre et al., 1982; Oscarsson and Rönnmark, 1989, 1990; Oscarsson, 1994; Oscarsson et al., 2001; Santolik and Parrot, 1996). The analysis needs the field data (either electric or magnetic field) and the dispersion relation for the linear Vlasov theory such as the WHAMP code (linear Vlasov dispersion solver) (Rönnmark, 1982, 50   1983).

The CSD matrix is constructed from the electric field measurements in the frequency domain. The CSD matrix can on the other hand be modeled as a projection of the wave polarization matrix $a_{ij}$ multiplied by the wave power in the wavevector domain over the dispersion relations $\omega_0(\boldsymbol{k})$. The number of waves does not need to be specified and the CSD matrix is modeled as an integral of the continuous wave energy distribution over the wavevectors by extracting the frequencies for the linear mode 55   waves using the Dirac delta function.

$$
\begin{aligned}
R_{ij}(\omega) &= \langle E_i E_j^* \rangle & (12)\\
&= \int \mathrm{d}^3 k \, a_{ij}(\boldsymbol{k}) \, F(\boldsymbol{k}) \, \delta(\omega - \omega_0(\boldsymbol{k})) & (13)
\end{aligned}
$$

### 2.4 Multi-probe method

Multi-spacecraft methods can be applied to multi-probe data analysis such as the timing analysis to measure the phase speed or the wave telescope technique or k-filtering technique to measure the fluctuation energy in the wavevector-frequency domain (Neubauer and Glassmeier, 1990; Pinçon and Lefeuvre, 1991; Motschmann et al., 1996; Glassmeier et al., 2001). The accessible dimension and the range of the wavevector domain are determined by the sensor configuration: 1-D wavevector domain for two probes in the direction of the probe alignment; 2-D domain if the probes are in the spin plane; and 3-D domain if the probes are 65   in the spin plane and along the spin axis. The advantages of using multiple probes are on the wavelength resolution such that the





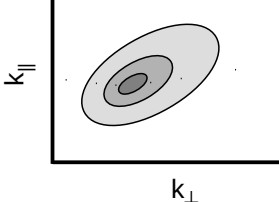

**Figure 5.** Wave distribution function $F(\boldsymbol{k})$.

fluctuation amplitudes are estimated directly in the wavenumber-frequency domain without using an assumption of dispersion relation or Taylor's frozen-in flow hypothesis. Moreover, the fluctuation amplitude obtained from the wave telescope technique retains the phase information and the wave-wave couplings can be studied in the wavevector domain, too. The highest accessible wavenumber is determined by the probe distance (spatial sampling distance), and the lowest recognizable wavenumber is about

10 to 50 times smaller than the highest wavenumber (Sahraoui et al., 2010). Upon the multi-probe data analysis, the spatial aliasing (Narita and Glassmeier, 2009) must be taken into account.

## 3  Turbulent fluctuations

### 3.1  Taylor's frozen-in flow hypothesis

Frequencies in the observer's (or spacecraft) frame are a sum of the intrinsic wave frequency, modulation of the intrinsic

frequency due to the random sweeping effect by the large-scale flow velocity fluctuation and the non-linear (sideband) effect, and the Doppler shift by the mean flow:

$$\omega = \omega_0 + \delta\omega + \boldsymbol{k} \cdot \boldsymbol{U}_0. \tag{14}$$

The Doppler shift dominates in a high-speed (super-sonic or super-Alfvénic) stream such that the frequencies can be mapped onto the wavenumbers along the flow (called the streamwise wavenumbers),

$\omega \simeq k_{\mathrm{fl}} U_0. \tag{15}$

The mapping is referred to as Taylor's frozen-in flow hypothesis (Taylor, 1938).

### 3.2  Stokes parameters

Using the Stokes parameters (Stokes, 1852; Berry et al., 1977) the perpendicular fluctuating magnetic fields (to the mean magnetic field) can be decomposed into a set of circularly polarized waves (Fig. 6). To do this, the perpendicular fluctuation

fields $B_{\perp 1}$ and $B_{\perp 2}$ must be expressed as complex numbers, and are obtained with the help of the Hilbert transform, $\hat{B}_{\perp 1}$ and





$\hat{B}_{\perp 2}$:

$$B'_{\perp 1} = B_{\perp 1} + \mathrm{i}\hat{B}_{\perp 1} \tag{16}$$

$$B'_{\perp 2} = B_{\perp 2} + \mathrm{i}\hat{B}_{\perp 2}. \tag{17}$$

Here the primed quantities $B'_{\perp 1}$ and $B'_{\perp 2}$ are the complex numbers. Two of the Stokes parameters, $I$ and $V$, are then deter-
mined:

$$I = \langle |B'_{\perp 1}|^2 \rangle + \langle |B'_{\perp 2}|^2 \rangle = \langle |B'_{\mathrm{R}}|^2 \rangle + \langle |B'_{\mathrm{L}}|^2 \rangle \tag{18}$$

$$V = -2\mathrm{Im}\left(\langle B'_{\perp 1}(B'_{\perp 2})^* \rangle\right) = \langle |B'_{\mathrm{R}}|^2 \rangle - \langle |B'_{\mathrm{L}}|^2 \rangle. \tag{19}$$

In this method, the Stokes $I$ parameter represents the total fluctuation energy in the perpendicular plane, and the Stokes $V$ the difference between the energy for the right-hand polarized fluctuations and that for the left-hand polarized. The energy for the
circularly rotating fields are obtained from $I$ and $V$ as

$$E_{\mathrm{R}} = \frac{I+V}{2} \tag{20}$$

$$E_{\mathrm{L}} = \frac{I-V}{2}, \tag{21}$$

for right-hand and left-hand rotation senses, respectively (Comişel et al., 2016).

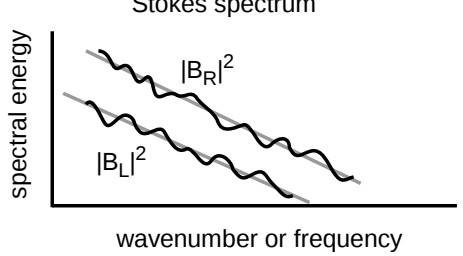

**Figure 6.** Stokes spectrum decomposing the fluctuations into circularly rotating fields.

### 3.3 Elsässer variables

The Elsässer variables are additive couplings of the magnetic field with the flow velocity by adapting the dimension of the magnetic field into that of the velocity:

$$z^{\pm} = \frac{\boldsymbol{B}}{\sqrt{\mu_0 \rho_0}} \pm \boldsymbol{U}, \tag{22}$$

where $\mu_0$ and $\rho_0$ denote the permeability of free space and the mass density, respectively.

The Elsässer variables give an intuitive picture of magnetohydrodynamics that parallel propagating Alfvén waves (to the
mean magnetic field) are expressed by $z^-$ because the fluctuating fields satisfy the relation $\boldsymbol{U} = -\boldsymbol{B}/\sqrt{\mu_0 \rho_0}$ (anti-correlation),


and anti-parallel propagating Alfvén waves by $z^+$ satisfying the correlation $\boldsymbol{U} = \boldsymbol{B}/\sqrt{\mu_0\rho_0}$. The nonlinear terms of the magnetohydrodynamic equation represent a coupling of a parallel propagating wave with an anti-parallel propagating wave, $z^-\nabla z^+$, and vice versa (Biskamp, 2003). The energy spectra can be estimated for $z^+$ and $z^-$, respectively (Fig. 7).

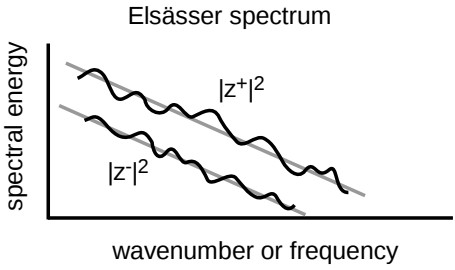

**Figure 7.** Elsässer variable spectrum.

### 3.4 Helicity quantities

Helicity quantities play an important role in turbulence. Magnetic helicity and cross helicity are invariant in ideal magnetohydrodynamics, and can even cascade onto different spatial scales. Current helicity and kinetic helicity are also useful quantities.

The magnetic helicity density is defined using the vector potential $\boldsymbol{A}$ and the magnetic field $\boldsymbol{B}$ as $h_{\mathrm{m}} = \langle \boldsymbol{A} \cdot \boldsymbol{B} \rangle$, and can be evaluated using the wavevector components and the off-diagonal elements of the CSD matrix for the magnetic field $R_{ij} = \langle B_i B_j^\dagger \rangle$ (Fig. 8 left panel). :

$$
\begin{aligned}
\quad h_{\mathrm{m}} \;=\; & -\frac{\mathrm{i}}{k^2}\left[k_{\mathrm{x}}(R_{\mathrm{yz}} - R_{\mathrm{zy}}) + k_{\mathrm{y}}(R_{\mathrm{zx}} - R_{\mathrm{xz}})\right. \\
& \left. + k_{\mathrm{z}}(R_{\mathrm{xy}} - R_{\mathrm{yx}})\right].
\end{aligned}
\tag{23}
$$

The vector potential is obtained by un-curling the equation $\boldsymbol{B} = \nabla \times \boldsymbol{A}$ and applying the Coulomb gauge $\nabla \cdot \boldsymbol{A} = 0$. The kinetic helicity density is obtained from the off-diagonal elements of the CSD matrix for the flow velocity, as well (Fig. 8 middle panel).

The cross helicity density represents a covariance between the flow velocity and the magnetic field, $h_{\mathrm{c}} = \langle \boldsymbol{U} \cdot \boldsymbol{B} \rangle$, which is the trace of the CSD matrix from the flow velocity and the magnetic field (Fig. 8 right panel). The cross helicity density can also be expressed as a difference between the two Elsässer variables, $h_{\mathrm{c}} = |z^+|^2 - |z^-|^2$ (Biskamp, 2003). The off-diagonal elements of the CSD matrix $\langle \boldsymbol{U}\boldsymbol{B}^\dagger \rangle$ contain the information on the electromotive force, amplifying the magnetic field in the dynamo mechanism. For single-point measurements, only streamwise wavenumbers can be determined using Taylor's hypothesis.


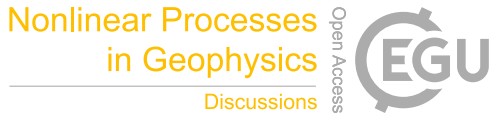

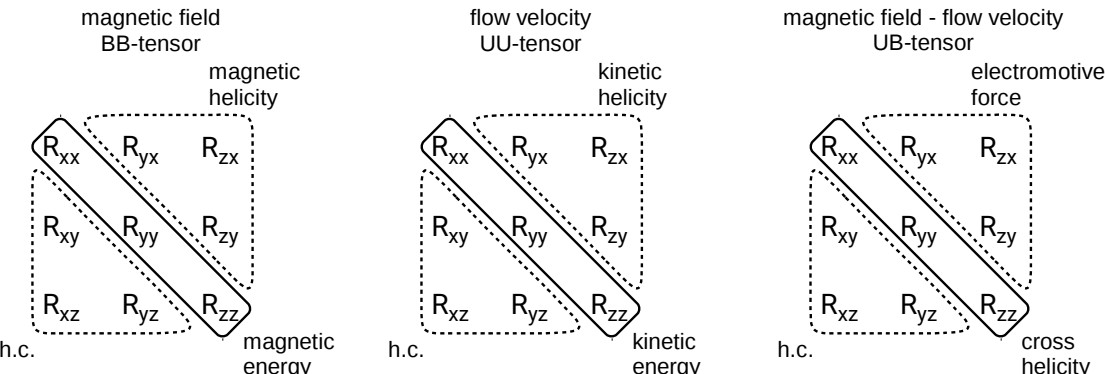

**Figure 8.** Structure of cross spectral density matrix for magnetic field and flow velocity data.

## 4 Wave-wave and wave-particle interactions

### 4.1 Higher-order statistics

Wave-wave couplings can be measured by extending the notion of covariance to multiple wave components. A three-wave coupling, for example, occurs under the condition of frequency and wavenumber conservations:

$$\omega_1 = \omega_2 + \omega_3 \qquad (24)$$

$$\boldsymbol{k}_1 = \boldsymbol{k}_2 + \boldsymbol{k}_3 \qquad (25)$$

The bispectrum estimator is a measure of three-wave couplings, and is constructed as a three-body covariance (Kim and Powers, 1979). The decay instability of a field-aligned large-amplitude Alfvén wave results in a backward-propagating Alfvén wave and a forward-propagating density fluctuation (sound wave), and is one of the leading mechanisms for a plasma to develop into turbulence (Longtin and Sonnerup, 1986; Terasawa et al., 1986; Wong and Goldstein, 1986; Hoshino and Goldstein, 1989) (see Fig. 9 right panel). The decay instability can be detected by constructing the bispectrum using the magnetic field fluctuation $B$ and the density fluctuation $n$ as

$$C^{(3)} = \left\langle B^*(\omega_1, k_{\parallel 1}) B(\omega_2, k_{\parallel 2}) n(\omega_3, k_{\parallel 3}) \right\rangle \qquad (26)$$

The bispectrum is non-zero if the resonance condition (Eqs. 24–25 including that for the initial phases) is satisfied. This fact can be seen by modeling the wave components as follows:

$$B(\omega_1, k_{\parallel 1}) = B_1 e^{i(\omega_1 t - k_{\parallel 1} z + \phi_1)} + \delta B \qquad (27)$$

$$B(\omega_2, k_{\parallel 2}) = B_2 e^{i(\omega_2 t - k_{\parallel 2} z + \phi_2)} + \delta B \qquad (28)$$

$$n(\omega_3, k_{\parallel 3}) = n_3 e^{i(\omega_3 t - k_{\parallel 3} z + \phi_3)} + \delta n \qquad (29)$$





Here $\delta B$ and $\delta n$ are random fluctuations or noise which vanishes after a statistical averaging, $\langle \delta B \rangle = 0$ and $\langle \delta n \rangle = 0$. Another application of three-wave couplings is spectral energy transport perpendicular to the mean magnetic field when the Alfvén wave dispersion relation is imposed (Biskamp, 2003) (Fig. 9 left panel). In the single spacecraft measurements, the bispectrum is studied either in the frequency domain or in the streamwise wavenumber domain.

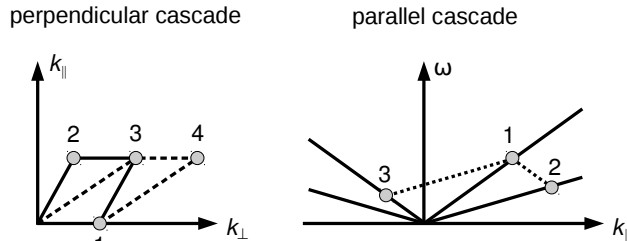

**Figure 9.** Three-wave coupling diagram for Alfvén wave scattering (left) and parametric decay of Alfvén waves into Alfvén and sound waves (right). Figure adapted from Narita (2012).

### 4.2 Landau and cyclotron resonances

Charged particles can exchange the energy with the wave electric field both parallel to the mean magnetic field (Landau resonance) and perpendicular to the mean field (Landau resonance). Fig. 10 left and middle panels show these resonance types and the associated parts of the distribution function schematically. The resonance parameters for the Landau and cyclotron resonances are:

$$\zeta_{\mathrm{L}} = \frac{\omega}{k_{\parallel} v_{(\mathrm{s})\mathrm{th}}}. \tag{30}$$

$$\zeta^{\mathrm{C}} = \frac{\omega - \Omega_{\mathrm{s}}}{k_{\parallel} v_{(\mathrm{s})\mathrm{th}}}. \tag{31}$$

Here the frequency $\omega$ is measured in the plasma rest frame. $\Omega_{\mathrm{s}}$ denotes the cyclotron frequency of particle species $s$ (ion species and electrons), $k_{\parallel}$ the parallel component of the wavevector, and $v_{(\mathrm{s})\mathrm{th}}$ the particle thermal speed of species $s$. In general, the resonance parameter can be defined for arbitrary harmonics of the cyclotron frequency ($m = 0, \pm 1, \pm 2, \cdots$):

$$\zeta^{(m)} = \frac{\omega - m\Omega_{\mathrm{s}}}{k_{\parallel} v_{(\mathrm{s})\mathrm{th}}}. \tag{32}$$

Note that the resonance parameters above are defined for a Maxwellian plasma. A correction is needed when treating a non-Maxwellian plasma to find the suitable velocity-space gradient for the resonance. The resonance is efficient when the parameter $\zeta_{\mathrm{L}}$ or $\zeta_{\mathrm{C}}$ is of the order of unity. Strictly speaking, the wave damping (or particle acceleration) is most efficient typically for $1 < \zeta < 5$. The upper limit (5 in this case) is not exact but the resonance becomes gradually inefficient at larger values of $\zeta$.





For $\zeta < 1$ the particle motion is slower than the wave propagation and the particles do not have a sufficient time for exchanging the energy with the wave electric field. For $\zeta > 5$ there are increasingly fewer particles with higher velocities for the resonance (higher than the thermal speed)

### 4.3 Pitch angle scattering

10 Charged particles can be scattered by the wave electric and magnetic fields incoherently, and the scattering deforms the velocity distribution function along the co-centric contours centered at the wave phase speed (Fig. 10 right panel). The reason for the deformation is that the particle kinetic energy $K_{\mathrm{wv}}$ (per unit mass) does not change in the co-moving frame with the apparent wave phase speed in the parallel direction to the mean magnetic field.

$$K_{\mathrm{wv}} = \frac{1}{2}\left[ v_\perp^2 + \left( v_\parallel - \frac{\omega}{k_\parallel} \right)^2 \right] = \text{const.} \tag{33}$$

15 The co-centric deformation of the distribution function achieves a quasi-linear equilibrium in that the velocity-space gradient becomes zero (plateau formation) in the pitch angle directions. The pitch angle scattering analysis was successful performed on the Helios ion data in the solar wind (Marsch, 2006; Marsch and Bourouaine, 2011), and obliquely propagating Alfvén/ion cyclotron waves are found to be the resonating waves. Note that the relevant phase speed is $\omega/k_\parallel$, and is different from the true phase speed $\omega/k$. The perpendicular component of the wavevector $k_\perp$ does not play a role in pitch angle scattering.

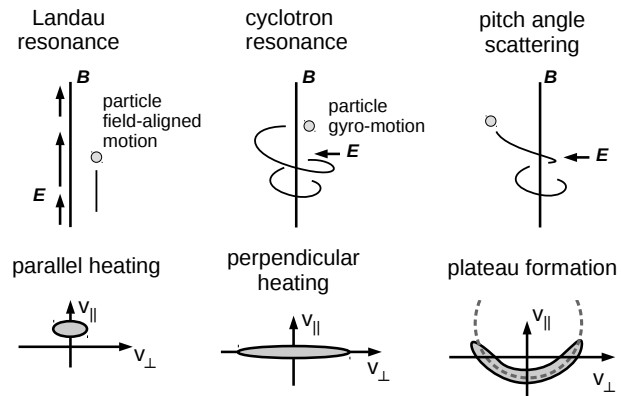

**Figure 10.** Wave-particle interactions and the associated part of the velocity distribution functions.

### 5 Outlook

20 Revealing the fluctuation properties is essential to advance our knowledge on turbulent plasmas using spacecraft data. In the following, particularly challenging questions are addressed that should be focused on for the upcoming spacecraft missions.



1. "What is the role of dispersion relation in turbulence?"

   Whether a dispersion relation exists in a turbulent field is a very important problem to guide to a theory of turbulence. One of the pictures of turbulence development is a transition from linear mode waves into more randomized nonlinear waves through the breakdown of the dispersion relation. The analysis of dispersion relation diagram is possible both from single-spacecraft and multi-spacecraft data. Perhaps the appearance of linear mode waves depends on the evolution time from the instability onset or the fluctuation amplitudes.

2. "What are the intrinsic spectra of turbulence?"

   Turbulence is essentially a spatially and temporally developing phenomenon, and the energy spectra must be viewed as a four-dimensional quantity as a function of the frequencies and the three components of the wavevectors. Turbulent fluctuations appear in the magnetic field, the electric field, and the plasma fluctuations such as the flow velocity, the density, and the temperature. Moreover, the magnetic and electric fields are vectorial quantities and a more complete picture of the spectra need to be constructed using the CSD matrices. As a consequence, a large number of spectra are necessary to characterize the turbulent fields unambiguously.

3. "How random are the wave phases in turbulence?"

   Turbulent fields cannot have fully random phases, since otherwise the constituent waves (or fluctuations) cannot interact with one another and the energy cascade through wave-wave interactions becomes impossible. The elementary energy transport process can occur only as a coherent process under the resonance conditions for the frequencies and the wavevectors. On the other hand, turbulent fluctuations are apparently incoherent. Otherwise the superposition of individual waves ends up with a large-scale coherent structure. The cascading waves generated by the wave-wave coupling attain more random phase at some stage of evolution.

Wave analysis methods assume that the measured fluctuation data are cleaned against noise or spacecraft-generated disturbance by proper calibration procedures. The signals in the data must clearly identified to separate from the noise. In the case of magnetometer data, the offsets and the noise floor must be determined prior to the data analysis.

Turbulent fields may contain coherent structures with eddies, current sheets, and discontinuities. Coherent structures can be conveniently studied by, for example, introducing the de Hoffmann-Teller frame for the shock waves (eliminating the convective electric field with the sliding frame along with the plane of the discontinuity), the analysis of electrostatic potential through the Liouville mapping, or visualization of the magnetic field and the plasma distribution using the Grad-Shafranov equation for a magnetohydrodynamic quasi-equilibrium state.

The fluctuations can also be highly intermittent such that the small-scale burst-like fluctuations are more localized and the fluctuation statistics strongly deviate from a Gaussian process. Various methods have been developed to characterize the intermittency such as the probability density function, the local intermittency measure, the multi-fractal method, the partition function, and the partial variance increments, wave phase shuffling and surrogation.





55   *Acknowledgements.*  This work is financially supported by Austrian Space Applications Programme at Austrian Research Promotion Agency, FFG ASAP-12 SOPHIE (Solar Orbiter wave observation program in the heliosphere) under contract 853994. The author acknowledges discussions with the Science Study Team for the ESA THOR mission concept.



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
