# Peer review of "Review article: Wave analysis methods for space plasma experiment"

_Nonlinear Processes in Geophysics, 2016_

## Referee Comment (RC1) · Anonymous Referee #1 · 19 Jan 2017

This paper provides a useful overview on data analysis methods applied to the study of monochromatic modes, turbulence and resonance/scattering phenomena. This kind of review represents something that was missing in the present Literature and as such deserves to be published after a short revision, as suggested below. However, this review would gain additional value if the Author would add some paragraphs on relatively recent techniques, wavelets based, allowing to define the "mean field direction" and the parallel and perpendicular directions to it, scale by scale (see Horbury et al., 2008, He et al., 2011, Telloni and Bruno, 2016, among others). This is not a criticism but rather a suggestion that the Author is free to take it or leave it since it does not change my overall positive evaluation of this review.

List of points to be revised:

1) ln 14: specify that these values refer to 1 AU

2) pg 3, ln 52: better to replace "that" with "where" or "such that"

3) pg 4, Figure 1: would it be possible to add the ion-cyclotron branch in Figure 1? Incidentally, the Author reports the following paper: Marsch, E., and Tu, C. Y., Evidence for pitch-angle diffusion of solar wind protons in resonance with ion-cyclotron waves, J. Geophys. Res., 106, 8357–8361, 2001, in the reference list but he never recalls this paper throughout the text.

4) pg 4, ln 35: it would be useful to add a reference to Bruno and Carbone, LNP, 928, 2016 where coherent structures in turbulence are treated extensively.

5) pg 5, matrix definition (3): this looks more like the definition of Auto-Spectral Density Matrix rather than Cross-Spectral Density Matrix since the general signal is only one: B(t) The CSD is usually intended to be obtained from the cross correlation matrix of two different signals, say A(t) and B(t).

6) pg 6, ln 76: Please, add reference to Bavassano & Bruno, 94, 11977, 1989

7) pg 6, ln 77: Could the Author be more explicit when he mentions that the "Rotation sense of the field fluctuation is evaluated from the off-diagonal elements of the CSD matrix." ? Otherwise add a reference to Arthur et al., 1976.

8) pg 12, ln 30: add a reference to Matthaeus and Goldstein, this was the first estimate of these invariants in the solar wind.

9) pg 12, ln 42: add a reference to add Tu and Marsch 1995

10) pg 14, ln 64: add a reference to Shebalin et al. (1983)

11) pg 16, ln 44: insert "be" after "must"

---

## Referee Comment (RC2) · Anonymous Referee #2 · 31 Jan 2017

The work is about a review of analysis methods commonly used in solar wind turbulence. The manuscript is nicely written, concise and surely appropriate for Nonlinear processes in Geophysics. The paper can be published almost in its present form. However, the quality of the paper can be improved taking into account the following comments.

1) Page 4, section about coherent structures (lines 31-36) The role of coherent structures such as current sheets and possible associated mechanisms such as magnetic reconnection should be further highlighted. This is a big topic for the community, since these structures are ubiquitous in the free solar wind as well as in magnetospheric plasma. In this regard, it would be very instructive to mention:

[] A. Greco, W. H. Matthaeus, S. Servidio, P. Chuychai, and P. Dmitruk, "Statistical

analysis of discontinuities in solar wind ACE data and comparison with intermittent MHD turbulence", The Astrophysical Journal Letters 691, L111 (2009).

Note that these structures populate signals at very high-cadence, on scales on the order of the electron skin depth, playing a role in the low frequency fluctuations (omega~0). Recently, this issue has been investigated in

[] A. Greco, S. Perri, S. Servidio, E. Yordanova, P. Veltri, "he complex structure of magnetic field discontinuities in the turbulent solar wind", The Astrophysical Journal Letters 823 (2), L39 (2016)

2) Page 5, equation (3) It would be more clear for the reader if the dependenc of "R" as a function of omega dependence is explicitly reported. Namley "R -> R(omega)".

3) Page 5, line 55, sentence: "by chopping the time interval into sub-intervals and averaging the matrix over the sub-intervals."

This "chopping" procedure, essentially, should have a more profound meaning. The ensemble averages, as in equation (2), consist of a large number of realizations, over several correlation length-scales (or correlation times), and over different experiments (solar wind dataset). This deals with the ergodic theorem, which is crucial in every turbulence measurement (see for example classic lecture notes and books on hydro-dynamics). "Chopping" the data at very small scale, unfortunately, violates this ensemble average, leading to ephemeral results. Unfortunately this habit became today a classical analysis technique. Although I do not agree with these methods, it would be important for the reader to (at least) know the problem of the "violation of ergodicity".

4) Page 5, lines 66-68 It would be nice to mention here some of the works made by Tim Horbury and colleagues on the definition of local mean field. Together with this, note that the definition of local mean field and its interpretation in the framework of plasma turbulence has been questioned in:

[] W. H. Matthaeus et al., "Local anisotropy, higher order statistics, and turbulence

spectra", The Astrophysical Journal 750 (2), 103 (2012)

5) Page 10, equation 14. It would be very interesting to spend more words about "\delta omega", which is crucial for the sweeping effect and therefore for the Taylor hypothesis.

6) Eq. 23, page 12 It is important here to mention the first work about the measurement of magnetic helicity in the solar wind, namely

[] W. H. Matthaeus, M. L. Goldstein, "Measurement of the rugged invariants of magnetohydrodynamic turbulence in the solar wind", Journal of Geophysical Research, 87 (A8), 6011 (1982)

---

## Author Comment (AC1) · 12 Feb 2017

Thank you very much for the comments. The manuscript revision is attached as a supplementary material.

*"This paper provides a useful overview on data analysis methods applied to the study of monochromatic modes, turbulence and resonance/scattering phenomena. This kind of review represents something that was missing in the present Literature and as such deserves to be published after a short revision, as suggested below. However, this review would gain additional value if the Author would add some paragraphs on relatively recent techniques, wavelets based, allowing to define the "mean field direction" and the parallel and perpendicular directions to it, scale by scale (see Horbury et al., 2008, He et al., 2011, Telloni and Bruno, 2016, among others). This is not a criticism but rather*

*a suggestion that the Author is free to take it or leave it since it does not change my overall positive evaluation of this review."*

A paragraph was added about the mean field determination (page 8, line 17 to page 9, line 2, starting with "Different approaches are possible to determine...")  with the references to Horbury et al. (2008), Wicks et al. (2010, 2011), Chen et al. (2011), He et al. (2011), Telloni and Bruno (2016).

*"List of points to be revised:"*

1. *"1) ln 14: specify that these values refer to 1 AU"*

   Done. (page 1, line 14).

2. *"2) pg 3, ln 52: better to replace "that" with "where" or "such that""*

   Yes. "such that" (page 3, line 14).

3. *"3) pg 4, Figure 1: would it be possible to add the ion-cyclotron branch in Figure 1? Incidentally, the Author reports the following paper: Marsch, E., and Tu, C. Y., Evidence for pitch-angle diffusion of solar wind protons in resonance with ion-cyclotron waves, J. Geophys. Res., 106, 8357–8361, 2001, in the reference list but he never recalls this paper throughout the text."*

   The ion-cyclotron branch was added to Fig. 1 and I indicate the electron-cyclotron and the ion-cyclotron branches in gray. The text was modified accordingly (page 3, lines 26–30; page 4, figure 1 caption). The reference to Marsch and Tu (2001) was added. (page 16, line 16)

4. *"4) pg 4, ln 35: it would be useful to add a reference to Bruno and Carbone, LNP, 928, 2016 where coherent structures in turbulence are treated extensively."*

Done. (page 5, line 2)

5. *"5) pg 5, matrix definition (3): this looks more like the definition of Auto-Spectral Density Matrix rather than Cross-Spectral Density Matrix since the general signal is only one: B(t). The CSD is usually intended to be obtained from the cross correlation matrix of two different signals, say A(t) and B(t)."*

I deleted the word "cross" and simply say "the spectral density matrix" (page 5, line 15) to avoid a confusion between what is "cross" and what is "auto" (for example in figure 8).

6. *"6) pg 6, ln 76: Please, add reference to Bavassano & Bruno, 94, 11977, 1989"*

Done. (page 7, line 9)

7. *"7) pg 6, ln 77: Could the Author be more explicit when he mentions that the "Rotation sense of the field fluctuation is evaluated from the off-diagonal elements of the CSD matrix." ? Otherwise add a reference to Arthur et al., 1976."*

The meaning of rotation sense is already explained on page 7, line 22 ("The value of ..."). The equations to determine the ellipticity are refered to in the beginning of the section with a reference to Fowler et al. (1967) and Arthur (1976). (page 7, lines 10–11)

8. *"8) pg 12, ln 30: add a reference to Matthaeus and Goldstein, this was the first estimate of these invariants in the solar wind."*

[Figure]

Done. (page 13, line 8)

9. *"9) pg 12, ln 42: add a reference to add Tu and Marsch 1995"*

Done. (page 13, line 21)

10. *"10) pg 14, ln 64: add a reference to Shebalin et al. (1983)"*

Done. (page 15, line 8)

11. *"11) pg 16, ln 44: insert 'be' after 'must'"*

Done. (page 17, line 25)

Please also note the supplement to this comment:
http://www.nonlin-processes-geophys-discuss.net/npg-2016-77/npg-2016-77-AC1-supplement.pdf

**Supplement:**

[revised manuscript text omitted]

---

## Author Comment (AC2) · 12 Feb 2017

Thank you very much for the comments. The manuscript revision is attached as a supplementary material.

*"The work is about a review of analysis methods commonly used in solar wind turbulence. The manuscript is nicely written, concise and surely appropriate for Nonlinear processes in Geophysics. The paper can be published almost in its present form. However, the quality of the paper can be improved taking into account the following comments."*

1. *"Page 4, section about coherent structures (lines 31-36) The role of coherent structures such as current sheets and possible associated mechanisms such as*

*magnetic reconnection should be further highlighted. This is a big topic for the community, since these structures are ubiquitous in the free solar wind as well as in magnetospheric plasma. In this regard, it would be very instructive to mention:*

- *A. Greco, W. H. Matthaeus, S. Servidio, P. Chuychai, and P. Dmitruk, "Statistical analysis of discontinuities in solar wind ACE data and comparison with intermittent MHD turbulence", The Astrophysical Journal Letters 691, L111 (2009).*
  *Note that these structures populate signals at very high-cadence, on scales on the order of the electron skin depth, playing a role in the low frequency fluctuations (omega $sim$ 0). Recently, this issue has been investigated in*
- *A. Greco, S. Perri, S. Servidio, E. Yordanova, P. Veltri, "The complex structure of magnetic field discontinuities in the turbulent solar wind", The Astrophysical Journal Letters 823 (2), L39 (2016)"*

Done. (page 4, line 16 to page 5, line 2)

2. *"Page 5, equation (3) It would be more clear for the reader if the dependenc of "R" as a function of omega dependence is explicitly reported. Namley "R -> R(omega)". "*

Done. I added the dependence on the frequency omega in Eq. (2) and (3).

3. *"Page 5, line 55, sentence: "by chopping the time interval into sub-intervals and averaging the matrix over the sub-intervals."*
   *This "chopping" procedure, essentially, should have a more profound meaning. The ensemble averages, as in equation (2), consist of a large number of realizations, over several correlation length-scales (or correlation times), and over different experiments (solar wind dataset). This deals with the ergodic theorem,*

*which is crucial in every turbulence measurement (see for example classic lecture notes and books on hydro- dynamics). "Chopping" the data at very small scale, unfortunately, violates this ensem- ble average, leading to ephemeral results. Unfortunately this habit became today a classical analysis technique. Although I do not agree with these methods, it would be important for the reader to (at least) know the problem of the "violation of ergodicity". "*

This is a very important remark. I added two paragraphs, one about the implementaion (classical method, page 6, lines 1–9) and the other about the problem with the chopping as a caveat (page 6, lines 10–15).

4. *"Page 5, lines 66-68 It would be nice to mention here some of the works made by Tim Horbury and colleagues on the definition of local mean field. Together with this, note that the definition of local mean field and its interpretation in the framework of plasma turbulence has been questioned in:*

   - *W. H. Matthaeus et al., "Local anisotropy, higher order statistics, and turbulence spectra", The Astrophysical Journal 750 (2), 103 (2012)"*

Done. (page 9, line 2)

5. *"Page 10, equation 14. It would be very interesting to spend more words about "delta omega", which is crucial for the sweeping effect and therefore for the Taylor hypothesis."*

I added the following sentences.

The random sweeping is a representation of turbulent fluctuations that small-scale fluctuations are swept by large-scale flow variation (Kraichnan 1964). The

large-scale variation (or the sweeping velocity) can be modeled to be random, Gaussian, and independent from the small-scale turbulent fluctuations. (page 11, lines 15–17)

6. *"Eq. 23, page 12 It is important here to mention the first work about the measurement of magnetic helicity in the solar wind, namely*

   - *W. H. Matthaeus, M. L. Goldstein, "Measurement of the rugged invariants of magnetohydrodynamic turbulence in the solar wind", Journal of Geophysical Research, 87 (A8), 6011 (1982)"*

   Done. (page 13, lines 16–17)

Please also note the supplement to this comment:
http://www.nonlin-processes-geophys-discuss.net/npg-2016-77/npg-2016-77-AC2-supplement.pdf